# Effects of Puffing, Acid, and High Hydrostatic Pressure Treatments on Ginsenoside Profile and Antioxidant Capacity of Mountain-Cultivated *Panax ginseng*

**DOI:** 10.3390/foods12112174

**Published:** 2023-05-28

**Authors:** Jang-Hwan Kim, Jae-Sung Shin, Wooki Kim, Hyungjae Lee, Moo-Yeol Baik

**Affiliations:** 1Department of Food Science and Biotechnology, Institute of Life Science and Resources, Kyung Hee University, Yongin 17104, Republic of Korea; zongy5000@naver.com (J.-H.K.); drumlover@naver.com (J.-S.S.); kimw@khu.ac.kr (W.K.); 2Department of Food Engineering, Dankook University, Cheonan 31116, Republic of Korea

**Keywords:** mountain-cultivated *Panax ginseng*, puffing, acid, high hydrostatic pressure, ginsenosides, antioxidant capacity

## Abstract

The purpose of this study was to investigate the effects of puffing, acid, and high hydrostatic pressure (HHP) treatments on the ginsenoside profile and antioxidant capacity of mountain-cultivated *Panax ginseng* (MCPG) before and after treatments. Puffing and HHP treatments decreased extraction yield and increased crude saponin content. The combination of puffing and HHP treatment showed significantly higher crude saponin content than each single treatment. Puffing treatment showed the highest ginsenoside conversion compared with HHP and acid treatments. Significant ginsenoside conversion was not observed in HHP treatment but was in acid treatment. When the puffing and acid treatments were combined, Rg3 and compound K content (1.31 mg and 10.25 mg) was significantly higher than that of the control (0.13 mg and 0.16 mg) and acid treatment (0.27 mg and 0.76 mg). No synergistic effect was observed between acid and HHP treatments. In the case of functional properties, the puffing treatment showed a significant increase in TFC (29.6%), TPC (1072%), and DPPH radical scavenging capacity (2132.9%) compared to the control, while acid and HHP combined treatments did not significantly increase; therefore, the synergistic effects of HHP/puffing and acid/puffing treatments were observed in crude saponin content and ginsenoside conversion, respectively. Consequently, puffing combined with acid or HHP treatments may provide new ways to produce high-value-added MCPG with a higher content of Rg3 and compound K or crude saponin compared to untreated MCPG.

## 1. Introduction

*Panax ginseng* has been known to have various physiological effects such as fatigue recovery, immunity enhancement, anticancer, and anti-inflammation, and has been used for the treatment of diseases and health promotion over thousands of years [1,2,3]. The various biological effects of ginseng are attributed to phenolic compounds, flavonoid content, acidic polysaccharides, and saponins. The contents of the functional compounds and physicochemical properties of ginseng are affected by the cultivation environment [4]. Farm-grown ginseng has problems such as residual fertilizer and pesticide used for cultivation. For these reasons, the demand for wild ginseng is increasing and mountain-cultivated *Panax ginseng* (MCPG), which is grown without artificial control after planting ginseng seeds in the mountain area, can be an alternative to wild *Panax ginseng*.

Ginsenosides, unique saponin components of ginseng, have a glycoside structure and exhibit various effects depending on structural differences. Major ginsenosides can be converted to minor ones by various processes [5,6,7]. Recently, studies on minor ginsenosides including Rg3 and compound K have been actively performed; Rg3 is a representative protopanaxatriol (PPT)-type minor ginsenoside, while compound K is a non-natural ginsenoside produced by biotransformation from protopanaxadiol (PPD)-type ginsenosides [8], and they have anticancer, anti-inflammatory, antiobesity, and neuroprotective effects [9,10,11,12]. Due to the excellent pharmaceutical activities of minor ginsenosides, much research has been conducted to convert major ginsenosides to minor ones through physical and chemical techniques [13]. Additionally, ginseng polysaccharide has been reported to have anticancer and immune control effects [14,15]. The polysaccharide of ginseng, known to have anticancer and immunoregulatory effects, is composed of neutral polysaccharides and acidic polysaccharides [16,17]. Acidic polysaccharide is a pectin-like compound with a molecular mass of 15,000 or higher, and its main components are acidic sugars such as galacturonic acid, glucuronic acid, and mannuronic acid. Acidic polysaccharide is known to have a higher level of effect on the immune system than neutral polysaccharides [14,15]. The chemical properties of polysaccharides in ginseng are known to be changed in the process of manufacturing red ginseng. Through this chemical change, red ginseng has a higher acidic polysaccharide content than white ginseng, and shows immune enhancement [17].

Red ginseng is a typical and representative heat-processed ginseng and has a high storage stability through steaming and drying processes. Red ginseng has been known to have a high level of minor ginsenosides, especially ginsenoside Rg3, due to the conversion of major ginsenosides by heat treatment. Organic acids in ginseng play an important role in the ginsenoside conversion process [18], and major ginsenosides could be converted into minor ones by acid treatment at room temperature [19]. Additionally, acid treatment has been reported to be a possible processing method for ginsenoside conversion [20]. Puffing causes the explosive evaporation of internal water using rapid pressure change to form a porous structure and increase the specific volume [21]. Puffing makes not only morphological changes but also physicochemical changes including starch gelatinization, protein denaturation, and enzyme inactivation [22]. It has been reported that major ginsenosides decreased and minor ginsenosides increased as puffing pressure increased when red ginseng was puffed [20]. High hydrostatic pressure (HHP) is a nonthermal process that affects the steric structure of a polymer substance by using a pressure of several hundred MPa. HHP has been known to cause starch gelatinization, protein denaturation, enzyme activation/inactivation, and pasteurization [23,24,25,26]. In addition, HHP induces the destruction of cell walls, resulting in increases in mass transfer rate and extraction rate [27,28]. Many studies have researched the effect of various processing on the physicochemical and physiological characteristics of ginseng, but limited information is available on the physicochemical and physiological characteristics of processed MCPG. Therefore, we attempted to investigate the single or combined effects of acid, HHP, and puffing treatments on the ginsenoside profile and antioxidant capacity of MCPG.

## 2. Materials and Methods

### 2.1. Materials

MCPG grown for four years on the Chung-Ok mountain (Pyeongchang, Republic of Korea) and harvested in September 2020 was obtained from Woori Do (Pyeongchang, Republic of Korea). The surface of the MCPG was washed three times under running tap water to remove soil. An additional washing of MCPG using distilled water was completed before the surface of the MCPG was dried. The dried sample was stored at −20 °C prior to use.

### 2.2. Chemicals

Ethanol was supplied by Ethanol Supplies World Co. (Jeonju, Republic of Korea). Diethyl ether, *n*-butanol, methyl alcohol, anhydrous ethyl alcohol (99.9%), sodium carbonate, sodium hydroxide, acetic acid, citric acid, hydrochloric acid, and oxalic acid were purchased from Daejung Chemicals & Metals Co. (Siheung, Republic of Korea). 2,2-Diphenyl-1-picrylhydrazyl (DPPH), Folin-Ciocalteu reagent, carbazole, galacturonic acid, gallic acid, and catechin were supplied by Sigma-Aldrich (St. Louis, MO, USA). Ascorbic acid was purchased from Reagent Duksan (Ansan, Republic of Korea). Aluminum chloride and sodium nitrite were obtained from Junsei Chemical (Tokyo, Japan). Ginsenosides standards such as Rg1, Re, Rf, Rg2, Rb1, Rc, Rb2, F1, Rd, F2, and Rg3 for HPLC analysis were supplied by Ambo Institute (Daejeon, Republic of Korea).

### 2.3. Puffing Treatment

The final moisture content of MCPG was adjusted to 14% (wet basis) at 40 °C in a hot air dryer (HB-502M; HanBeak Scientific Co., Bucheon, Republic of Korea). To prevent MCPG carbonization at high temperature, the mixture of rice with the dried MCPG at a ratio of 50:1 (*w*/*w*) was heated in a rotary gun puffing machine (SH-P10; PPsori Co., Namyangju, Republic of Korea). When the internal pressure reached 490 kPa, the pressure decreased to 294 kPa. Subsequently, the machine was heated again to elevate the pressure up to 980 kPa. Upon reaching this pressure, the chamber door of the machine was opened to puff MCPG. Puffing treatment was carried out three times.

### 2.4. HHP and Acid Treatment

In the case of fresh and puffed MCPG, 10 mL of distilled water was added to 1 g dried MCPG. For acid treatment, 10 mL of 0.5 M oxalic acid was added to 1 g dried MCPG. The mixture was ground using a blender and subsequently put in a polyethylene pouch. The air inside the pouch was removed as much as possible and sealed with a heat sealer (SK-310; TAMSTECH, Namyangju, Republic of Korea). In case of atmospheric acid treatment, a sample was treated at room temperature for 15 min. In case of HHP acid treatment (HR-S-1000; Ilshin Autoclave, Daejeon, Republic of Korea), a sample was treated at 550 MPa for 15 min. After the HHP treatment was completed, the sample was neutralized to pH 4.75 using 2 M NaOH. A sample with distilled water instead of both acid solutions was used as a control. HHP treatment was carried out three times.

### 2.5. Extraction Yield

The extraction of MCPG was mixed with 70% ethanol at room temperature for 30 min using a magnetic stirrer. A funnel with a Whatman No. 2 filter paper (GE Healthcare Life Sciences, Piscataway, NJ, USA) was loaded onto a Kimble filtering flask to filter the extracted MCPG. Subsequently, the filtrate as the final extract was dried at 105 °C in a hot-air dryer (HB-502M; HanBeak Scientific Co., Bucheon, Republic of Korea) until there was no change in weight. The extraction yield was calculated using the following equation:$$\mathrm{Extraction}\;\mathrm{yield}\;\left(\%\right)=\frac{\left(W_{2}-W_{1}\right)}{A}\times \frac{E}{\;E^{\prime }}\times 100$$

W_1_: weight of empty aluminum dish (g)

W_2_: weight of aluminum dish and solid (g)

A: weight of dried ginseng (g)

E: total volume of extract (mL)

E′: used volume of extract (mL)

The filtrate was concentrated using a rotary vacuum evaporator (N-11, EYELA, Tokyo, Japan). For further analyses, concentrated samples dissolved in distilled water were aliquoted.

### 2.6. Crude Saponin Content

The crude saponin content of the extract was measured according to the method of An et al. [20] with some modification. To determine the crude saponin content, 10 mL of concentrated MCPG was mixed with 15 mL of distilled water and 25 mL of diethyl ether in a separatory funnel. The mixture was agitated and subsequently left for 30 min to separate the water and the ether layers. Only the water layer was mixed thoroughly with 25 mL of water-saturated *n*-butanol by shaking, and then it was left for 30 min to allow the water and *n*-butanol layers to separate. The *n*-butanol layer was collected, and the water layer was mixed again with 25 mL water-saturated *n*-butanol two more times to collect the separated *n*-butanol layer repeatedly. Finally, 50 mL of distilled water was added to the collected water-saturated *n*-butanol layer. The mixture was agitated and left to stand until the water-saturated *n*-butanol layer had separated. The separated water-saturated *n*-butanol layer was concentrated under reduced pressure using a rotary evaporator. The resulting concentrate was dried at 105 °C for 2 h. The crude saponin content was calculated using the following equation:$$\mathrm{Crude}\;\mathrm{saponin}\;\mathrm{content}\;\left(\frac{\mathrm{mg}}{g\;\mathrm{dried}\;\mathrm{ginseng}}\right)=\frac{\left(W_{1}-W_{2}\right)}{W_{3}}\times \frac{A}{B}$$

W_1_: weight of the dried sample and flask (mg)

W_2_: weight of the flask (mg)

W_3_: weight of total dried ginseng (g)

A: weight of total concentrates (g)

B: weight of used concentrates (g)

### 2.7. Ginsenoside Profile

The ginsenoside profile was determined using the method of An et al. [20] with some modification. For HPLC analysis, the Agilent 1260 infinity II LC system (Agilent Technologies, Santa Clara, CA, USA) was equipped with a Kinetex C18 column (4.6 × 50 mm; Phenomenex, Torrance, CA, USA) and a UV detector at 203 nm. Crude saponin concentrate was dissolved in 5 mL of HPLC grade methanol and was filtered using a 0.45 µm Millipore filter as a HPLC injection sample. Five microliters of the sample were loaded onto the column at 45 °C. The mobile phase of the system consisted of distilled water (A) and acetonitrile (B) at a flow rate of 0.6 mL/min under the following conditions: 81% A and 19% B, 0–7 min; 71% A and 29% B, 7–14 min; 60% A and 40% B, 14–25 min; 44% A and 56% B, 25–28 min; 30% A and 70% B, 28–30 min; 10% A and 90% B, 30–31.5 min; 10% A and 90% B, 31.5–34 min; 81% A and 19% B, 34–34.5 min; 81% A and 19% B, 34.5–40 min. 

### 2.8. Functional Properties

#### 2.8.1. Total Flavonoid Content (TFC)

The total flavonoid content (TFC) was measured using the method of Zhishen et al. [28] with some modification. In brief, 0.5 mL of diluted extract was mixed with 3.2 mL of distilled water and 0.15 mL of 5% NaNO_2_. After 5 min, 0.15 mL of 10% AlCl₃ was added to the mixture. After 1 min of incubation, 1 mL of Na_2_CO_3_ was finally added. The absorbance of the mixture was measured at 510 nm (UV-1200; Labentech, Incheon, Republic of Korea). Catechin served as a standard, and the equation of the calibration curve was Y = 0.0028 X − 0.0116 (R^2^ = 0.997). TFC was expressed as mg catechin equivalent (CE)/g dried ginseng.

#### 2.8.2. Total Phenolic Content (TPC)

The total phenolic content (TPC) was estimated using the method of Singleton and Rossi [29] with some modification. Briefly, 200 µL of extract was mixed with 2.6 mL of distilled water and 200 µL of Folin & Ciocalteu’s solution and was left for 6 min, followed by the addition of 2 mL of 7% Na_2_CO_3_ to the mixture. The mixture was left to react at 23 °C for 90 min, and then the absorbance of the reaction mixture was monitored at 750 nm (UV-1200; Labentech). Gallic acid was selected as a standard, and the equation of the calibration curve was Y = 0.0038 X + 0.0047 (R^2^ = 0.999). TPC was expressed as mg gallic acid equivalent (GAE)/g dried ginseng.

#### 2.8.3. DPPH Radical Scavenging Capacity

The radical scavenging capacity of the extract was measured with a 0.1 mM DPPH solution prepared using 2,2-Diphenyl-1-picrylhydrazyl and 80% methanol [30,31]. The absorbance of DPPH solution was adjusted to 0.650 ± 0.020 at 517 nm using 80% methanol. The extract (0.05 mL) was mixed with 2.95 mL of the adjusted DPPH solution, and the mixture was allowed to react at 25 °C for 30 min in a dark room. Subsequently, the absorbance was measured at 517 nm (UV-1200; Labentech). Vitamin C was selected as a standard, and the equation of the calibration curve was Y = 0.0025 X − 0.0062 (R^2^ = 0.999). The radical scavenging capacity was determined as mg vitamin C equivalent (VCE)/g dried ginseng. 

### 2.9. Acidic Polysaccharides

The amount of acidic polysaccharide was measured using the method of Do et al. [32] with some modification. Five milliliters of the extract was mixed with 3 mL of distilled water and 0.25 mL of 0.1% carbazole. The mixture was placed in a water bath at 85 °C for 5 min. After cooling at room temperature, the absorbance was measured at 525 nm (UV-1200; Labentech). The amount of acidic polysaccharides was expressed as mg galacturonic acid equivalent (GalAE)/g dried ginseng.

### 2.10. Statistical Analysis

All the experiments were repeated three times and each measurement was triplicated. Experimental data were analyzed using Analysis of Variance (ANOVA) and expressed as mean value ± standard deviation. Duncan’s multiple range test was applied to assess significant differences among experimental data (*p* < 0.05). All the statistical analyses were carried out using SAS software (version 8.2, SAS Institute, Inc., Cary, NC, USA).

## 3. Results and Discussion

### 3.1. Extraction Yield and Crude Saponin Content

The extraction yield and crude saponin content of puffed, as well as acid and HHP treated MCPG, are shown in Figure 1. In the case of acid-treated samples, accurate measurement was impossible due to residual salts after neutralization. Therefore, only the results of the raw MAPG (control), puffed, and HHP-treated MCPG are shown. Single or combined treatments decreased the extraction yield in the order of control > puffing > HHP > puffing and HHP. Puffed (P-MCPG)- and HHP-treated MCPG (H-MCPG) showed an 11.19% and 18.99% decrease compared to the control, respectively. Puffed and HHP-treated MCPG (PH-MCPG) showed the lowest extraction yield with a 28.88% decrease compared to the control. The decrease in extraction yield by puffing was possibly due to the carbonization of sample. The MCPG has a thinner and a longer root shape and a round band shape as compared with farm-grown ginseng [33]. Because of this morphological characteristic, MCPG could cause more carbonation than farm-grown ginseng, resulting in a decrease in extraction yield. Moreover, HHP treatment decreased the extraction yield in this case, which is not consistent with a previous report [27]. It has been reported that the solubility of most natural compounds increases with increasing pressure, and high pressure may accelerate the rate of mass transfer, which enhances both solvent penetration and the release of intracellular molecules [27]. 

Unlike extraction yield, single or combined treatments increased the crude saponin content in the order of puffing and HHP > HHP > puffing > control. P-MCPG and H-MCPG showed a 5.29% and 11.31% increase, respectively. Puffing treatments have been reported to increase the dissolution rate of the internal components by destroying the cell wall and forming a porous structure [5,6,20]. In addition, HHP treatments have been reported to increase the mass transfer rate by destroying the cell wall [27]. These phenomena might cause a synergistic increase in crude saponin after combined treatments of puffing and HHP, resulting in a 26.78% increase, which was significantly higher than the sum of P-MPGP and H-MCPG. 

### 3.2. Ginsenoside Profile

HPLC chromatograms of MCPG are shown in Figure 2 and Figure 3. Raw MCPG (control) showed higher levels of major ginsenosides such as Rg1, Rb1, and Rb2 and lower levels of minor ginsenosids such as Rg3 and compound K (Figure 2A). H-MCPG showed a similar tendency in HPLC chromatogram to the control (Figure 2B). In the case of acid-treated MCPG (A-MCPG), the content of major ginsenosides decreased, while minor ginsenosides such as Rg3 and compound K peaks slightly increased (Figure 2C). In the case of the combined treatments of HHP and acid treatments (HA-MCPG), its ginsenoside profile was similar to that of A-MCPG (Figure 2D). These results indicate that ginsenoside conversion in the nonthermal condition was not greatly affected by HHP treatment but was greatly influenced by acid treatment, and consequently, there is no synergistic effect between acid and HHP treatments. In contrast, the HPLC chromatogram of P-MCPG showed only minor ginsenoside peaks and almost no major ginsenoside peaks (Figure 3A). P-MCPG showed significant increases in the levels of Rg3 and compound K compared to the control. Puffed and acid-treated MCPG (PA-MCPG), PH-MCPG, and the combination of the three treatments (PHA-MCPG) resulted in similar ginsenoside profiles to that of P-MCPG (Figure 3B–D). 

The ginsenoside content of MCPG according to each treatment is shown in Table 1. H-MCPG showed slightly higher ginsenoside content than the control, possibly due to the high amount of crude saponin, which likely came from the breakdown of cell walls and the increase in the mass transfer after HHP treatment [27]. In the case of A-MCPG, the content of major ginsenosides such as Rg1, Re, Rf, Rg2, Rb1, Rc, and Rb2 decreased, while the content of minor ginsenosides such as Rg3, Rh2, and compound K increased compared to those of the control and H-MCPG, possibly due to the acidic deglycosylation and conversion of major ginsenosides to minor ones [12,19]. HA-MCPG showed a similar ginsenoside profile to that of A-MCPG, suggesting that there was no synergistic effect from the addition of HHP to acid treatment. On the other hand, most of the major ginsenosides were not detected in P-MCPG; however, the content of minor ginsenosides such as Rg3 and compound K greatly increased, indicating that puffing treatment may cause a great ginsenoside conversion. There was no significant difference when HHP treatment was added, suggesting that HHP treatment did not show the synergistic effect seen in the combined treatments of acid and HHP. Unlike the addition of HHP treatment, acid treatment showed a synergistic effect with the puffing treatment. PA-MCPG showed a significantly higher content of ginsenoside Rg3 and compound K than P-MCPG; this result was probably due to the additional ginsenoside conversion of intermediate products produced by puffing. Major ginsenosides were known to be converted to ginsenoside Rg3 and compound K through several steps [34]. Thus, various intermediate products might be present after puffing. These intermediate products could be converted to ginsenoside Rg3 and compound K through additional acid treatment, which could result in a high ginsenoside Rg3 and compound K content. The ginsenoside profile of PHA-MCPG was not significantly different compared to that of PA-MCPG, implying no synergistic effect of HHP on ginsenoside conversion after puffing and acid treatments. The ginsenoside conversion was reported to improve with increasing puffing pressure [35]. However, there is a limitation in puffing because side effects such as carbonization occur at high pressure and temperature. Therefore, the combined treatment of puffing/acid treatments might be applied to the efficient processing of MCPG for higher ginsenoside conversion without carbonization.

### 3.3. Functional Properties

The functional properties determined by TFC, TPC, and DPPH radical scavenging capacity are shown in Figure 4. Only the puffing treatment increased the TFC, and single acid or HHP treatments or combined treatments decreased the TFC (Figure 4A). H-MCPG, A-MCPG, and HA-MCPG showed 37.7%, 52.3%, and 70.4% decreases compared to the control, respectively. Although P-MCPG showed a 29.6% increase, PH-MCPG, PA-MCPG, and PAH-MCPG showed 1.31%, 1.81%, and 6.4% decreases compared to the control. Overall, acid and HHP treatments decreased TFC in both nonpuffed and puffed MCPG, suggesting that flavonoids might be degraded or disintegrated by acid and HHP treatments. TPC, DPPH radical scavenging capacity, and acidic polysaccharide increased with puffing as well as acid treatments (Figure 4B–D) with a similar pattern. Without puffing, acid treatment significantly increased both TPC and DPPH radical scavenging capacity, but with puffing, acid treatment significantly increased the TPC but not in the case of DPPH radical scavenging capacity and acidic polysaccharide. HHP treatment did not affect the TPC, DPPH radical scavenging capacity, or acidic polysaccharide in both puffed and nonpuffed MCPG.

Puffing showed an overwhelming increase in TPC, DPPH radical scavenging capacity, and acidic polysaccharide compared with the other treatments. P-MCPG showed a 1071.8% increase in TPC, a 2132.9% increase in DPPH radical scavenging capacity, and a 189% increase in acidic polysaccharide; these were not greatly decreased after combined acid and HHP treatments. These results were attributable to the formation of porous structure and the chemical bond-breaking by the puffing process [6], and could be explained by the increase in certain substances such as maltol, a Maillard reaction, and the increase in dissolution rate of antioxidants after puffing [36,37,38]. Similarly, it has been reported that TFC, TPC, and DPPH radical scavenging capacity in wild *Panax ginseng* were increased by puffing [5]. The decrease or no change in TFC, TPC, and DPPH radical scavenging capacity after HHP could be attributed to the decrease in the extraction yield.

## 4. Conclusions

This study investigated the effects of acid, HHP, and puffing treatments and the synergistic effects of combined treatments on the ginsenoside profile and antioxidant capacity of MCPG. Both HHP and puffing processes reduced extraction yields by 18.99% and 11.19%, respectively, and a synergistic effect between HHP and puffing on extraction yield was not found. In contrast, the crude saponin content increased by 11.31% and 5.29% through HHP and puffing, respectively. In addition, the combined process showed a 26.78% increase in crude saponin content, indicating the synergistic effect between HHP and puffing. Although HHP treatment did not significantly affect the ginsenoside profile, acid and puffing treatments converted major ginsenosides to minor ones in MCPG. Furthermore, the combined treatment of acid and puffing resulted in 1.31 mg and 10.25 mg of ginsenoside Rg3 and compound K, respectively, which was significantly higher than those of the control (0.13 mg and 0.16 mg). This result suggests that the intermediate ginsenosides formed during puffing could be converted to Rg3 or compound K with additional acid treatment. Only the puffing treatment increased the antioxidant capacity and acidic polysaccharide. Synergistic effects between acid, HHP, and puffing were not observed. This study confirmed that single or combined acid, puffing, and HHP treatments have their own characteristic effects on the physicochemical properties of MCPG. Synergistic effects were observed in crude saponin content (puffing and HHP treatments) and ginsenoside conversion (puffing and acid treatments). This work confirmed that combined physicochemical treatments can produce high-functional MCPG containing larger amounts of ginsenoside Rg3 as well as compound K, and it is expected to provide a new processing method for high-value-added MCPG.

## Figures and Tables

**Figure 1 foods-12-02174-f001:**
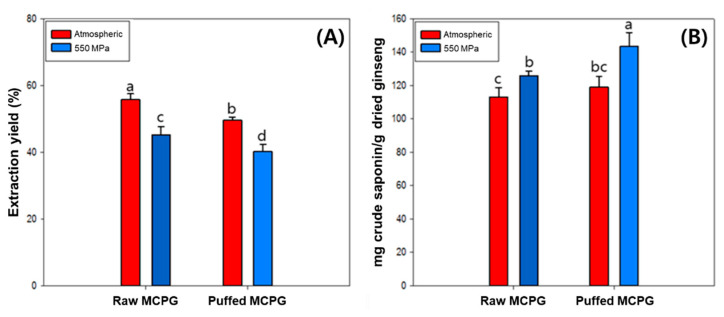
Extraction yield (**A**) and crude saponin content (**B**) of MCPG with HHP and puffing treatments. Different letters above the bars indicate significant differences *(p <* 0.05).

**Figure 2 foods-12-02174-f002:**
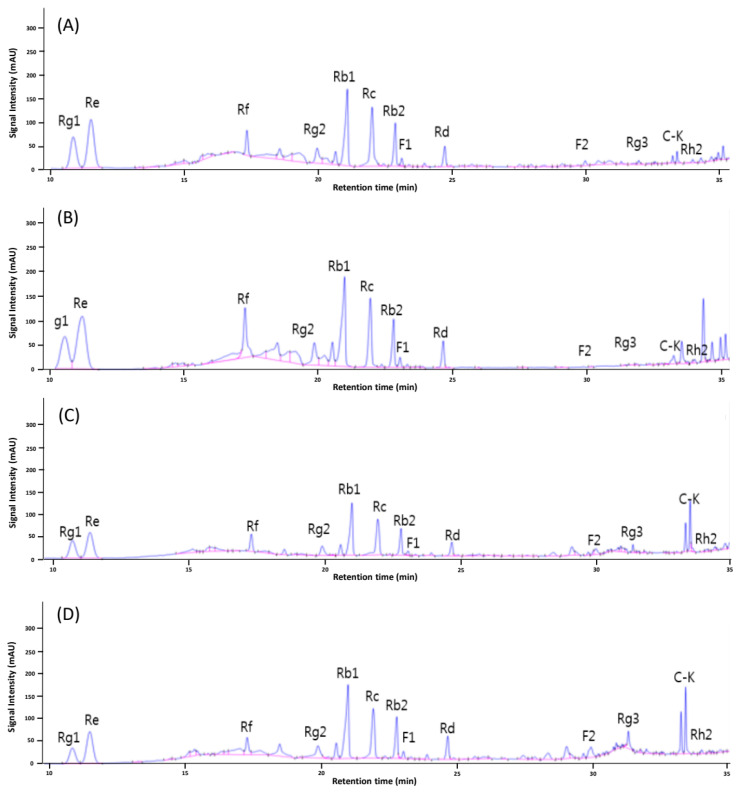
HPLC chromatograms of MCPG with acid and HHP treatments. (**A**) Raw MCPG (control), (**B**) H-MCPG, (**C**) A-MCPG, (**D**) HA-MCPG. C-K: compound K.

**Figure 3 foods-12-02174-f003:**
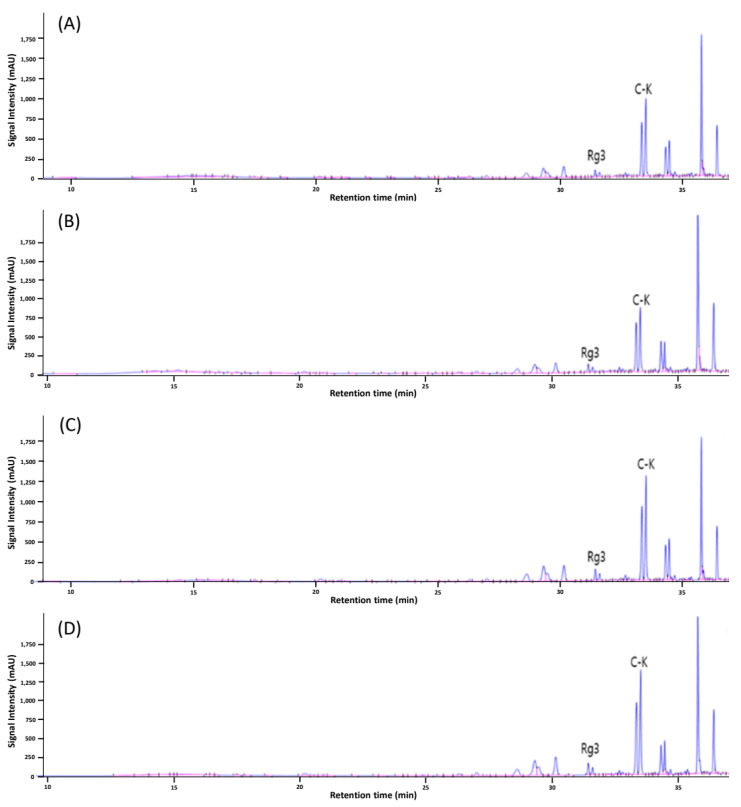
HPLC chromatograms of puffed MCPG with acid and HHP treatments. (**A**) P-MCPG, (**B**) PH-MCPG, (**C**) PA-MCPG, (**D**) PHA-MCPG. C-K: compound K.

**Figure 4 foods-12-02174-f004:**
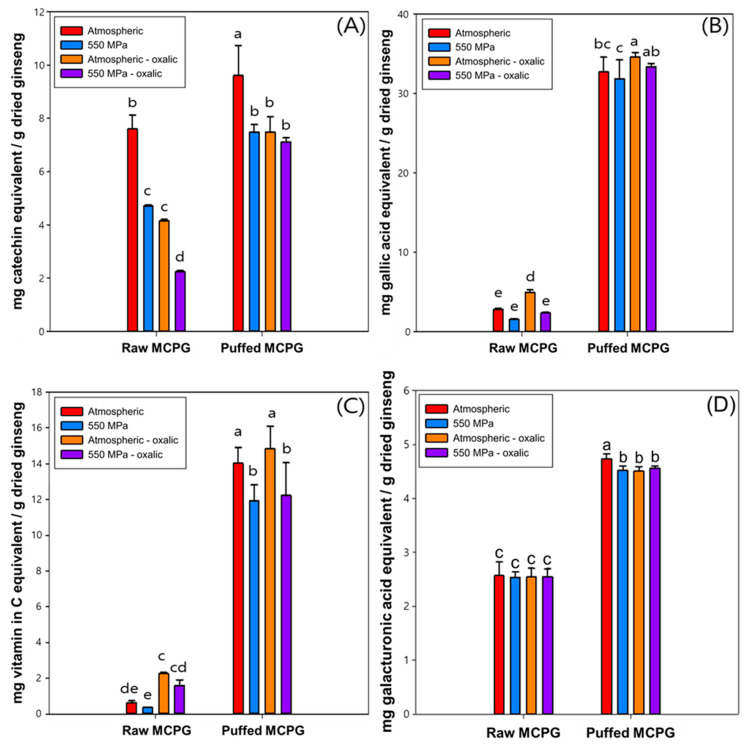
TFC (**A**), TPC (**B**), DPPH radical scavenging activity (**C**), and acidic polysaccharide of MCPG with acid, HHP, and puffing treatments (**D**). Different letters above the bars indicate significant differences *(p <* 0.05).

**Table 1 foods-12-02174-t001:** Amounts of ginsenosides in acid-, HHP-, and puffing-treated MCPG.

	Rg1	Re	Rf	Rg2	Rb1	Rc	Rb2	F1	Rd	F2	Rg3	C-K **	Rh2
Control	2.31 ± 0.29 ^B^ *	5.31 ± 0.89 ^B^	2.43 ± 0.35 ^AB^	0.97 ± 0.07 ^B^	5.29 ± 0.35 ^AB^	3.00 ± 0.79 ^A^	3.28 ± 0.13 ^A^	0.17 ± 0.01 ^D^	0.78 ± 0.01 ^AB^	0.04 ± 0.01 ^D^	0.13 ± 0.02 ^F^	0.16 ± 0.02 ^F^	0.06 ± 0.09 ^B^
H-MCPG	3.75 ± 0.06 ^A^	6.21 ± 0.26 ^A^	2.13 ± 0.40 ^AB^	1.11 ± 0.25 ^AB^	5.52 ± 0.45 ^A^	3.45 ± 0.28 ^A^	3.25 ± 0.10 ^A^	0.30 ± 0.30 ^C^	0.70 ± 0.08 ^C^	0.06 ± 0.02 ^BCD^	0.20 ± 0.02 ^E^	0.15 ± 0.01 ^F^	0.03 ± 0.00 ^B^
A-MCPG	1.49 ± 0.12 ^D^	3.23 ± 0.35 ^C^	1.63 ± 0.12 ^C^	0.66 ± 0.03 ^C^	4.77 ± 0.70 ^BC^	2.25 ± 0.31 ^B^	2.31 ± 0.50 ^B^	0.52 ± 0.04 ^A^	0.57 ± 0.02 ^B^	0.27 ± 0.00 ^A^	0.33 ± 0.03 ^D^	0.76 ± 0.06 ^E^	0.14 ± 0.01 ^A^
HA-MCPG	1.87 ± 0.10 ^C^	3.52 ± 0.28 ^C^	1.93 ± 0.17 ^BC^	1.22 ± 0.12 ^A^	4.52 ± 0.34 ^C^	3.21 ± 0.35 ^A^	1.90 ± 0.10 ^C^	0.38 ± 0.02 ^B^	0.85 ± 0.12 ^A^	0.06 ± 0.01 ^BCD^	0.36 ± 0.02 ^D^	0.93 ± 0.03 ^E^	0.02 ± 0.00 ^B^
P-MCPG	nd ***	0.05 ± 0.00 ^D^	0.29 ± 0.01 ^D^	0.58 ± 0.03 ^C^	0.10 ± 0.02 ^D^	0.05 ± 0.00 ^C^	0.09 ± 0.00 ^D^	nd ***	0.04 ± 0.01 ^F^	0.05 ± 0.00 ^CD^	0.70 ± 0.04 ^C^	6.44 ± 0.33 ^C^	0.04 ± 0.00 ^B^
PH-MCPG	nd ***	nd ***	0.3 ± 0.02 ^D^	0.63 ± 0.05 ^C^	0.05 ± 0.00 ^D^	0.05 ± 0.00 ^C^	0.11 ± 0.02 ^D^	nd ***	0.12 ± 0.02 ^EF^	0.04 ± 0.00 ^D^	0.81 ± 0.03 ^B^	5.84 ± 0.15 ^D^	0.06 ± 0.01 ^B^
PA-MCPG	nd ***	0.09 ± 0.00 ^D^	0.37 ± 0.02 ^D^	0.93 ± 0.03 ^B^	0.17 ± 0.01 ^D^	0.11 ± 0.00 ^C^	0.14 ± 0.01 ^D^	nd ***	0.13 ± 0.02 ^E^	0.07 ± 0.00 ^B^	1.31 ± 0.50 ^A^	10.25 ± 0.10 ^B^	0.04 ± 0.00 ^B^
PHA-MCPG	nd ***	0.06 ± 0.01 ^D^	0.36 ± 0.00 ^D^	0.95 ± 0.01 ^B^	0.10 ± 0.02 ^D^	0.09 ± 0.01 ^C^	0.11 ± 0.00 ^D^	0.02 ± 0.00 ^E^	0.25 ± 0.01 ^D^	0.06 ± 0.00 ^BC^	1.36 ± 0.02 ^A^	10.57 ± 0.25 ^A^	0.04 ± 0.00 ^B^

* Different letters within a column indicate significant differences *(p <* 0.05). ** C-K: compound K. *** nd: not detected.

## Data Availability

In this case, data is unavailable due to privacy or ethical restrictions.

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
