# Peer review of "Effects of Puffing, Acid, and High Hydrostatic Pressure Treatments on Ginsenoside Profile and Antioxidant Capacity of Mountain-Cultivated Panax ginseng"

_foods, 2023, doi:10.3390/foods12112174_

Round 1

Reviewer 1 Report

Some deficiencies are observed. For example,

Abstract mentions "The puffing treatment showed the highest ginsenoside conversion", when is high?

Introduction has old references (1994) and does not present the relevance of the research.

Methodology. Authors should use international units sistems, be homogeneous in words, Ej. TPC (total phenolic compounds or compound). There are formulas, not well written. (NaNO).

Results. These are not compartive with other documents, therefore, these are not justified.

References. These are not well cited, authors should not use et al.,

Check some words "running water"

Author Response

Response to reviewer 1

Some deficiencies are observed. For example,

Abstract mentions "The puffing treatment showed the highest ginsenoside conversion", when is high?

(Ans) Thank you for your comment. As mentioned in abstract, puffing performed at 980 kPa only and puffing treatment revealed the highest ginsenoside conversion compared to HHP and acid treatments. We changed it for better understanding. Please see line 21-22 in the revised manuscript.

Introduction has old references (1994) and does not present the relevance of the research.

(Ans) Thank you for your comment. The two old references reported the immunomodulatory effect and pharmacological activity of ginseng. Therefore, we cited those references. However, we removed the old references and added two recent references with the relavance of this research as you suggested. Please see line 354-358 in the revised manuscript.

Methodology. Authors should use international units systems, be homogeneous in words, Ej. TPC (total phenolic compounds or compound). There are formulas, not well written. (NaNO).

(Ans) Thank you for your comment. We carefully checked and corrected accordingly. All changes were marked in red in the revised manuscript.

Results. These are not comparative with other documents, therefore, these are not justified.

(Ans) Thank you for your comment. Unfortunately, there are not many researches in the effects of three treatments on ginsenoside profile and antioxidant activity of mountain cultivated Panax Ginseng. However, we tried to find a single treatment effect on ginsenoside profile and antioxidant activity of ginseng and compared. All changes were marked in red in the revised manuscript.

References. These are not well cited, authors should not use et al.,

(Ans) Thank you for your comment. We carefully checked and corrected accordingly. All changes were marked in red in the revised manuscript.

Reviewer 2 Report

The article presented for review concerns the effects of puffing, acid and high hydrostatic pressure  (HHP) treatments on ginsenoside profile and antioxidant activity of mountain cultivated panax  ginseng (MCPG). This article requires many revisions. In general, the article lacks information on the possibility of using the discussed processes in food production, nutrition technology, etc. It is necessary to demonstrate and emphasize the possibilities, legitimacy, etc. of using the research material subjected to the discussed processes in food in general. This aspect, in my opinion, is essential, given the nature of Foods magazine. Below are the main comments.

1.       In the case of the title of the article, I suggest not including the abbreviation MCPG, which tells little to the reader, but the full Latin name.

2.       In the "Abstract" section:

·       Line 15 The purpose of this study was .. (not is)

·       Line 17-18 “HHP was treated at 550 MPa and puffing was treated at 980kPa. For acidic 17 treatment, 0.5 M oxalic acid was used and neutralized to pH 4.75 using 2 M NaOH” - this should be in the methodological part rather than in the abstract

·       Line 21-23 - combine into one sentence

·       Line 24-25 “No synergistic effect was observed between the acid and 24 the HHP treatments” - So is it worth emphasizing in the abstract?

·       Line 26 “HHP treatments resulted in insignificant change” - This is an incorrect expression or inference (check throughout the manuscript as this form of inference and statement appears many times).

·       The most important - Quantitative results are missing - please add !!!

·       The abstract should contain the key information, the most important and the most valuable results. At the moment - it's hard to say what the dependencies were. A value is required. Moreover, the authors keep repeating that processes caused an increase, a decrease, that "something" was important and something was not important?? Without numerical values, the abstract adds nothing. Therefore, the use of phrases, quote "Acidic polysaccharides also showed the same tendency" (Line 26-27) is unfounded - it is not known what tendencies are meant?

·       Similarly, in the case of the main conclusion in this abstract, i.e. lines 27-29. In order to draw such a "conclusion", one should give specific numerical values, and not just repeat laconic phrases that something "is important" or "is unimportant".

·       Please rephrase this abstract.

3.       In the "Introduction" section:

·       The botanical taxonomy and characteristics of Panax ginseng should be detailed, further the economic value of Panax ginseng in agri-food industry should be paid

·       Numerous stylistic errors (not only in the content of the introduction) often make it difficult to understand the content of the statement. For example, line 37 - the various effects of ginseng are not due to the content of flavonoids, but rather their biological activity.... Please correct the English language throughout the manuscript.

·       Line 46 – What kond of effects?? Please Please clarify

·       Line 47-48 “Recently, studies on minor ginsenosides 47 have been actively conducted” - Citation is missing

·       Line 48 - Rg3 - Authors should explain the abbreviation and write anything more about these components, i.e. about Rg3 and compound K

·       Line 50 - There is a period missing at the end of the sentence

·       Line 54 – “Furthermore, it was found that major ginsenosides could be converted into 53 minor ginsenosides by acid treatment at room temperature.” - Citation is missing

·       Line 64 “HHP has effects such as sterilization, starch  gelatinization and protein denaturation, enzyme activation and inactivation” – Please, provide a bit more detail.

·       Line 68 – “Therefore, the purpose of this study is to investigate…” not “is” but “was”

·       Line 45 – 67 - The authors refer to the research of other authors, however, no results and no values ​​appear in this section. Please add the results of research by other authors.

4.       In the "Materials and Methods" section:

·       Line 73- 74 – There are no details regarding the origin of the research material, its origin, cultivation conditions, region (the term "specific region" is insufficient to characterize the research material!), the date of collection of the number of collected and tested samples (this information is necessary to determine the representativeness of the tested material samples).

·       In how many repetitions were the tests performed? Because the information is only in section 2.9. "Statistical analysis", but the information contained therein concerns the number of repetitions of the determination, not the number of samples of the research material. Please complete the missing information.

·       Line 77-85 - Perhaps section 2.2 should be separated. "chemicals"?

·       Line 87 - There is no information about the research equipment ("hot air dryer") - please complete

·       Line 89 - There is no information about the research equipment ("puffing machine") - please complete

·       Line 95, “0.5 M and 10 mL” - Do not start a sentence with numerical values

·       Line 98 - There is no information about the research equipment ("heat sealer") - please complete

·       Line 99 - There is no information about the research equipment ("In case of HHP-acid treatment"…) - please complete

·       Line 107 – “and dried at 105°C” - How long?

·       Line 120-139 - Is the method of determining the content of saponins an original method?? Is it a methodology known in the literature, modified by the authors?

·       Line 140-148 -  Is the method of determining the ginsenoside profile an original method?? Is it a methodology known in the literature, modified by the authors?

·       Line 143 - There is no information about the research equipment ("II LC system"…) - please complete

·       Line 154, 160, 167 - The research equipment (spectrophotometer) information is missing, please comlete

·       Please provide equations of calibration curves for standard substances, ie for catechin (line 155), gallic acid (line 161).

·       Line 165 “0.05 mL” - Do not start a sentence with numerical values

·       Line 177 “3mL” -  Do not start a sentence with numerical values

5.       In the "Results and Discussion" section:

·       Line 190 – typo, not “saaponin” but saponin

·       Please change the descriptions of graph 1, graph 4 and graph 5 to the appropriate font (it is also worth adjusting its size, especially on the vertical axis), also in table 1 the values ​​are written in a different font - please correct.

·       Discussion and discussion of the obtained results requires improvement and supplementation. First, the authors tend to compare the significance and irrelevance of the results. They do not compare the results themselves. For example, line 220 "was not significantly affected by pressure", line 221 "there was no significant synergistic effect", line 222-223 "a significant increase in Rg3 and compound K peaks." The RESULTS and trends should be discussed, and the paper should not be emphasized (note to the further part of the discussion) that the authors obtained: "no significant synergistic effect was observed .." (Line 229), "There was no significant difference" (Line 231), "There was no significant difference between" (Line 237), etc.

·       Line 273-276 - A bit more details should be added, the potential reader would be interested in changes in the profile of various bioactive compounds (and not only) under the influence of the processes used by the Authors.

·       Line 281-288 - This part should be in the "Introduction" section rather than in the "Results and Discussion" section.

·       Lines 289-294 - As already mentioned above, "Significance" or "lack of significance" is mentioned 5 times, but neither values nor information about the direction of the observed changes are mentioned.

·       In general, the presentation of the results obtained and their discussion are unsatisfactory and need improvement (as above).

·       When discussing the obtained results, one should use values ​​and not general "tendencies of change" because these tendencies are difficult to unambiguously "capture" from the text of the manuscript. Therefore, the results should be given and the tendencies of changes in the content of bioactive compounds under the influence of the applied processes should be emphasized.

·       The discussion needs to be expanded as there is no reference to any of these literature results in the current version of the manuscript. Comparative data is missing.

6.       In the "Conclusions" section:

·       Please add quantitative results

·       The novelty and applicability of the obtained research results should be emphasized

7.       In the "References" section:

·       Isn't there some more up-to-date literature? Most of the items are articles from before 2010

·       Minor errors in the list of references, for example item 17, item 24 - the year of publication should be written in bold

Numerous stylistic errors often make it difficult to understand the content of the statement.

Author Response

Response to Reviewer 2

The article presented for review concerns the effects of puffing, acid and high hydrostatic pressure  (HHP) treatments on ginsenoside profile and antioxidant activity of mountain cultivated panax  ginseng (MCPG). This article requires many revisions. In general, the article lacks information on the possibility of using the discussed processes in food production, nutrition technology, etc. It is necessary to demonstrate and emphasize the possibilities, legitimacy, etc. of using the research material subjected to the discussed processes in food in general. This aspect, in my opinion, is essential, given the nature of Foods magazine. Below are the main comments.

  1. In the case of the title of the article, I suggest not including the abbreviation MCPG, which tells little to the reader, but the full Latin name.

Ans) Thank you for your suggestion. We corrected it as you suggested. Please see title in the revised manuscript.

  1. In the "Abstract" section:
  • Line 15 The purpose of this study was .. (not is)
  • Line 17-18 “HHP was treated at 550 MPa and puffing was treated at 980kPa. For acidic 17 treatment, 0.5 M oxalic acid was used and neutralized to pH 4.75 using 2 M NaOH” - this should be in the methodological part rather than in the abstract
  • Line 21-23 - combine into one sentence
  • Line 24-25 “No synergistic effect was observed between the acid and 24 the HHP treatments” - So is it worth emphasizing in the abstract?
  • Line 26 “HHP treatments resulted in insignificant change” - This is an incorrect expression or inference (check throughout the manuscript as this form of inference and statement appears many times).
  • The most important - Quantitative results are missing - please add !!!
  • The abstract should contain the key information, the most important and the most valuable results. At the moment - it's hard to say what the dependencies were. A value is required. Moreover, the authors keep repeating that processes caused an increase, a decrease, that "something" was important and something was not important?? Without numerical values, the abstract adds nothing. Therefore, the use of phrases, quote "Acidic polysaccharides also showed the same tendency" (Line 26-27) is unfounded - it is not known what tendencies are meant?
  • Similarly, in the case of the main conclusion in this abstract, i.e. lines 27-29. In order to draw such a "conclusion", one should give specific numerical values, and not just repeat laconic phrases that something "is important" or "is unimportant".
  • Please rephrase this abstract.

Ans) Thank you for your suggestion. We corrected it as you suggested. Please see the abstract in the revised manuscript.

  1. In the "Introduction" section:
  • The botanical taxonomy and characteristics of Panax ginseng should be detailed, further the economic value of Panax ginseng in agri-food industry should be paid
  • Numerous stylistic errors (not only in the content of the introduction) often make it difficult to understand the content of the statement. For example, line 37 - the various effects of ginseng are not due to the content of flavonoids, but rather their biological activity.... Please correct the English language throughout the manuscript.
  • Line 46 – What kind of effects?? Please clarify
  • Line 47-48 “Recently, studies on minor ginsenosides 47 have been actively conducted” - Citation is missing
  • Line 48 - Rg3 - Authors should explain the abbreviation and write anything more about these components, i.e. about Rg3 and compound K
  • Line 50 - There is a period missing at the end of the sentence
  • Line 54 – “Furthermore, it was found that major ginsenosides could be converted into 53 minor ginsenosides by acid treatment at room temperature.” - Citation is missing
  • Line 64 “HHP has effects such as sterilization, starch  gelatinization and protein denaturation, enzyme activation and inactivation” – Please, provide a bit more detail.
  • Line 68 – “Therefore, the purpose of this study is to investigate…” not “is” but “was”
  • Line 45 – 67 - The authors refer to the research of other authors, however, no results and no values ​​appear in this section. Please add the results of research by other authors.

Ans) Thank you for your suggestion. We rewrote and rephrased the introduction as you suggested. Please see the introduction in the revised manuscript.

  1. In the "Materials and Methods" section:
  • Line 73- 74 – There are no details regarding the origin of the research material, its origin, cultivation conditions, region (the term "specific region" is insufficient to characterize the research material!), the date of collection of the number of collected and tested samples (this information is necessary to determine the representativeness of the tested material samples).

Ans) Thank you for your suggestion. We corrected it as you suggested. Please see line 85-86 in the revised manuscript.

  • In how many repetitions were the tests performed? Because the information is only in section 2.9. "Statistical analysis", but the information contained therein concerns the number of repetitions of the determination, not the number of samples of the research material. Please complete the missing information.

Ans) Thank you for your suggestion. We corrected it as you suggested. We added the repetition number in the revised manuscript. Please see lines 107-108 and 118 in the revised manuscript.

  • Line 77-85 - Perhaps section 2.2 should be separated. "chemicals"?

Ans) Thank you for your suggestion. We corrected it as you suggested. Please see line 90 in the revised manuscript.

  • Line 87 - There is no information about the research equipment ("hot air dryer") - please complete

Ans) Thank you for your suggestion. We added the information about the equipment as you suggested. Please see line 101 in the revised manuscript.

  • Line 89 - There is no information about the research equipment ("puffing machine") - please complete

Ans) Thank you for your suggestion. We added the information about the equipment as you suggested. Please see line 104 in the revised manuscript.

  • Line 95, “0.5 M and 10 mL” - Do not start a sentence with numerical values

Ans) Thank you for your suggestion. We corrected it as you suggested. Please see line 110-111 in the revised manuscript.

  • Line 98 - There is no information about the research equipment ("heat sealer") - please complete

Ans) Thank you for your suggestion. We added the information about the equipment as you suggested. Please see line 114 in the revised manuscript.

  • Line 99 - There is no information about the research equipment ("In case of HHP-acid treatment"…) - please complete

Ans) Thank you for your suggestion. We added the information about the equipment as you suggested. Please see line 115-116 in the revised manuscript.

  • Line 107 – “and dried at 105°C” - How long?

Ans) Thank you for your suggestion. We added the drying time as you suggested. Please see line 124 in the revised manuscript.

  • Line 120-139 - Is the method of determining the content of saponins an original method?? Is it a methodology known in the literature, modified by the authors?

Ans) Thank you for your suggestion. It is a modified method of An et al. (2011) and marked it in the revised manuscript. Please see line 138-139 in the revised manuscript. (An, Y.-E.; Ahn, S.-C.; Yang, D.-C.; Park, S.-J.; Kim, B.-Y.; Baik, M.-Y. Chemical conversion of ginsenosides in puffed red ginseng. LWT-Food Science and Technology 2011, 44, 370-374.)

  • Line 140-148 -  Is the method of determining the ginsenoside profile an original method?? Is it a methodology known in the literature, modified by the authors?

Ans) Thank you for your suggestion. It is a modified method of An et al. (2011) and marked it in the revised manuscript. Please see line 160 in the revised manuscript. (An, Y.-E.; Ahn, S.-C.; Yang, D.-C.; Park, S.-J.; Kim, B.-Y.; Baik, M.-Y. Chemical conversion of ginsenosides in puffed red ginseng. LWT-Food Science and Technology 2011, 44, 370-374.)

  • Line 143 - There is no information about the research equipment ("II LC system"…) - please complete

Ans) Thank you for your suggestion. We added the information about the equipment as you suggested. Please see line 161 in the revised manuscript.

  • Line 154, 160, 167 - The research equipment (spectrophotometer) information is missing, please complete

Ans) Thank you for your suggestion. We added the information about the equipment as you suggested. Please see lines 174-175, 181-182, 189-190 and 197 in the revised manuscript.

  • Please provide equations of calibration curves for standard substances, ie for catechin (line 155), gallic acid (line 161).

Ans) Thank you for your suggestion. We added the equations of calibration curves for standard substances as you suggested. Please see lines 175-176, 182-183 and 190-191 in the revised manuscript.

  • Line 165 “0.05 mL” - Do not start a sentence with numerical values

Ans) Thank you for your suggestion. We corrected it as you suggested. Please see line 187 in the revised manuscript.

  • Line 177 “3mL” -  Do not start a sentence with numerical values

Ans) Thank you for your suggestion. We corrected it as you suggested. Please see line 194 in the revised manuscript.

  1. In the "Results and Discussion" section:
  • Line 190 – typo, not “saaponin” but saponin

Ans) Thank you for your suggestion. We corrected it as you suggested. Please see line 207 in the revised manuscript.

  • Please change the descriptions of graph 1, graph 4 and graph 5 to the appropriate font (it is also worth adjusting its size, especially on the vertical axis), also in table 1 the values ​​are written in a different font - please correct.

Ans) Thank you for your suggestion. We corrected it as you suggested. Please see Fig.1, 4 and 5 as well as Table 1 in the revised manuscript.

  • Discussion and discussion of the obtained results requires improvement and supplementation. First, the authors tend to compare the significance and irrelevance of the results. They do not compare the results themselves. For example, line 220 "was not significantly affected by pressure", line 221 "there was no significant synergistic effect", line 222-223 "a significant increase in Rg3 and compound K peaks." The RESULTS and trends should be discussed, and the paper should not be emphasized (note to the further part of the discussion) that the authors obtained: "no significant synergistic effect was observed .." (Line 229), "There was no significant difference" (Line 231), "There was no significant difference between" (Line 237), etc.

Ans) Thank you for your suggestion. We rewrote, rephrased, and reorganized the results and discussions as you suggested.

  • Line 273-276 - A bit more details should be added, the potential reader would be interested in changes in the profile of various bioactive compounds (and not only) under the influence of the processes used by the Authors.

Ans) Thank you for your suggestion. We corrected it as you suggested. Please see line 262~267 in the revised manuscript.

  • Line 281-288 - This part should be in the "Introduction" section rather than in the "Results and Discussion" section

Ans) Thank you for your suggestion. We moved it to introduction as you suggested. Please see line 55~61 in the revised manuscript.

  • Lines 289-294 - As already mentioned above, "Significance" or "lack of significance" is mentioned 5 times, but neither values nor information about the direction of the observed changes are mentioned.

Ans) Thank you for your suggestion. We revised as you suggested. Please see line 285-297 in the revised manuscript.

  • In general, the presentation of the results obtained and their discussion are unsatisfactory and need improvement (as above).
  • When discussing the obtained results, one should use values ​​and not general "tendencies of change" because these tendencies are difficult to unambiguously "capture" from the text of the manuscript. Therefore, the results should be given and the tendencies of changes in the content of bioactive compounds under the influence of the applied processes should be emphasized.
  • The discussion needs to be expanded as there is no reference to any of these literature results in the current version of the manuscript. Comparative data is missing.

Ans) Thank you for your suggestion. We rewrote, rephrased, and reorganized the results and discussions as you suggested.

  1. In the "Conclusions" section:
  • Please add quantitative results
  • The novelty and applicability of the obtained research results should be emphasized

Ans) Thank you for your suggestion. We corrected it as you suggested. Please see line 315-333 in the revised manuscript.

  1. In the "References" section:
  • Isn't there some more up-to-date literature? Most of the items are articles from before 2010

Ans) Thank you for your suggestion. We added recent references as you suggested.

  • Minor errors in the list of references, for example item 17, item 24 - the year of publication should be written in bold

Ans) Thank you for your suggestion. We corrected it as you suggested.

Reviewer 3 Report

The manuscript submitted for review Effects of Puffing, Acid and High Hydrostatic Pressure 2 (HHP) Treatments on Ginsenoside Profile and 3 Antioxidant Activity of Mountain Cultivated Panax 4 ginseng (MCPG) by Jang-Hwan et al. is the original work. Everything is well designed, organized and written. I have only a slight remark about the methodology. Why do the authors use the % inhibition formula when the results are expressed in vitamin c equivalents. How was it calculated? And a second note about the antioxidant activity, I would use the equivalent of gallic acid, then the antioxidant activity can be correlated with TPC.

Author Response

Response to Reviewer 3

The manuscript submitted for review Effects of Puffing, Acid and High Hydrostatic Pressure 2 (HHP) Treatments on Ginsenoside Profile and 3 Antioxidant Activity of Mountain Cultivated Panax 4 ginseng (MCPG) by Jang-Hwan et al. is the original work. Everything is well designed, organized and written. I have only a slight remark about the methodology. Why do the authors use the % inhibition formula when the results are expressed in vitamin c equivalents. How was it calculated? And a second note about the antioxidant activity, I would use the equivalent of gallic acid, then the antioxidant activity can be correlated with TPC.

Ans) Thank you for your suggestion. We corrected them as you suggested. Please see materials and methods in the revised manuscript.

Round 2

Reviewer 1 Report

Thanks for making the observations, I recommend checking some observations that are still being observed.

Different letters in the names of the authors and different size (line 25). 

Line 125, 130. Use mL, h, respectively.

Line 152. The technical of flavonoids use NaNO2.

Lene 152. After 5 min,

Reviewer 2 Report

The Authors of the manuscript made corrections. In my opinion, the manuscript can be accepted for future stages of publication.